# An Evaluation of Parylene Thin Films to Prevent Encrustation for a Urinary Bladder Pressure MEMS Sensor System

**DOI:** 10.3390/polym15173559

**Published:** 2023-08-27

**Authors:** Sébastien Buchwalder, Mario Hersberger, Henrike Rebl, Susanne Seemann, Wolfgang Kram, Andreas Hogg, Lars G. W. Tvedt, Ingelin Clausen, Jürgen Burger

**Affiliations:** 1School of Biomedical and Precision Engineering, University of Bern, Güterstrasse 24/26, 3010 Bern, Switzerland; mario.hersberger@unibe.ch (M.H.); juergen.burger@med.unibe.ch (J.B.); 2Graduate School for Cellular and Biomedical Sciences, University of Bern, Mittelstrasse 43, 3012 Bern, Switzerland; 3Institute for Cell Biology, Rostock University Medical Center, Schillingallee 69, 18057 Rostock, Germany; henrike.rebl@med.uni-rostock.de (H.R.); susanne.seemann@med.uni-rostock.de (S.S.); 4Department of Urology, Rostock University Medical Center, Schillingallee 35, 18057 Rostock, Germany; wolfgang.kram@med.uni-rostock.de; 5Coat-X SA, Eplatures-Grise 17, 2300 La Chaux-de-Fonds, Switzerland; hogg@coat-x.com; 6InVivo Bionics AS, Gaustadallèen 21, 0349 Oslo, Norway; lars.g.w.tvedt@invivo.io (L.G.W.T.); ingelin.clausen@invivo.io (I.C.)

**Keywords:** MEMS encapsulation, urological implant, parylene, silicon oxide (SiO_x_), encrustation, zeta potential, wettability

## Abstract

Recent developments in urological implants have focused on preventive strategies to mitigate encrustation and biofilm formation. Parylene, a conformal, pinhole-free polymer coating, has gained attention due to its high biocompatibility and chemical resistance, excellent barrier properties, and low friction coefficient. This study aims to evaluate the effectiveness of parylene C in comparison to a parylene VT4 grade coating in preventing encrustation on a urinary bladder pressure MEMS sensor system. Additionally, silicon oxide (SiO_x_) applied as a finish coating was investigated for further improvements. An in vitro encrustation system mimicking natural urine flow was used to quantify the formation of urinary stones. These stones were subsequently analyzed using Fourier transform infrared spectrometry (FTIR). Encrustation results were then discussed in relation to coating surface chemical properties. Parylene C and VT4 grades demonstrated a very low encrustation mass, making them attractive options for encrustation prevention. The best performance was achieved after the addition of a hydrophilic SiO_x_ finish coating on parylene VT4 grade. Parylene-based encapsulation proved to be an outstanding solution to prevent encrustation for urological implants.

## 1. Introduction

In the 1930s, Dr. Frederick Foley introduced the indwelling urinary catheter to urologic practice, and these catheters remain widely used in daily urological procedures today [1]. Despite their long history of use, urinary catheters still present various disadvantages. A significant concern is the formation of crystalline biofilms that encrust and block the catheter. Encrustation can occur rapidly, leading to urine flow obstruction and potential complications, such as incontinence or painful bladder distention. These challenges highlight the need for improved solutions to minimize encrustation and enhance patient comfort and care [2,3].

Recently, various preventive surface modification strategies against encrustation and biofilm formation have been investigated. Most urinary stents consist of polymer mixtures with optimized surface properties designed to reduce encrustation. New polymeric stents, exhibiting optimized surface properties, are intended to improve biocompatibility and bio-dispersibility, ease the insertion and withdrawal, and reduce the encrustation [4]. To reduce mechanical friction and resistance during ureteral stent placement, hydrophilic substances, including hydrogel, polyvinylpyrrolidone (PVP), and polyvinylpyrrolidone-iodine (PVP-I) coated to the surface of the implant, can be used [5]. Laube et al. [6] compared polyurethane surfaces with hydrophilic amorphous carbon (a-C:H) surfaces. The experiment showed that crystalline biofilms on urological stents in artificial urine are strongly influenced by the formation and development of gas bubbles on the surface of the stents. Hydrophobic polyurethane surfaces allowed the formation of more extensive and stable bubbles than hydrophilic surfaces. Gas bubbles increase the deposition of salts from the artificial urine and are trapped as crystalline hollow spheres in the encrustation layer. In contrast, the bubbles from hydrophilic surfaces lift off quickly before the settling salts can trap them. Various strategies have been explored to mitigate encrustation and bacterial adherence in urinary tract implants. Kram et al. [3] investigated diamond-like carbon coatings combined with copper doping to kill bacterial growth while maintaining biocompatibility. Their findings indicated that increased surface roughness, due to copper doping, correlated with higher encrustation, underscoring the need for low-roughness anti-adhesive surfaces in future stent designs. Recent attention has shifted towards metal ion coatings (gold, silver, copper) for inherent bactericidal properties. Innovative approaches involving Ag–Au nanoparticle-coated stents offer prolonged antibacterial and biofilm-resistant qualities [7]. Additionally, Franz et al. [8] proposed a sandwich layer with a silver base layer and parylene film to gradually release non-toxic, bactericidal Ag ions. Early antimicrobial coating studies involved soaking urological stents in antibiotics and drying them for a coated effect; combining such coated stents with systemic antibiotics yielded bacteriostatic effects. Clarithromycin-coated stents combined with amikacin prevented biofilm infections, and rifampicin-coated stents with tigecycline inhibited biofilm formation [9,10].

From the medical point of view, monitoring pressure in biomedical applications is of the highest importance, with routine measurements involving blood, eye, urinary bladder, and joint pressures. To facilitate accurate in vivo pressure measurements, MEMS technology offers promising possibilities due to its small size, low weight, and low energy consumption [11]. However, exposure to aggressive body fluids and biological substances can lead to biofouling and alter device characteristics, compromising stability [12]. Traditional MEMS packaging methods, often involving bulky metal or solid polymer cages, limit the potential for miniaturized implantable MEMS. To prevent the body attacking the sensor, a thin biocompatible coating emerges as a viable alternative, providing sufficient protection while maintaining the advantage of small size, and not affecting the sensor’s functionalities. In response to these requirements, an in vivo sensor system was developed by Clausen et al. for direct pressure measurement into human organs [13,14]. The system comprises a small-sized and highly sensitive MEMS pressure sensor that is integrated into a catheter. An applied version of this sensor catheter has been developed to measure pressure inside the human bladder. It is wired to an external module responsible for biasing, sampling, converting, and storing sensor measurements. This innovative solution allows for precise physiological recordings during natural bladder filling and emptying cycles. By employing protective coatings, this sensor system offers a reliable and minimally invasive approach for urinary bladder monitoring, facilitating valuable insights for medical experts [15,16]. Clausen’s solution show how the integration of biocompatible coatings with MEMS devices presents new opportunities for developing miniaturized implantable medical sensors in various applications.

Poly-para-xylylene, commonly referred to as parylene and its derivates, is a thin-film highly conformal polymer coating often cited as the material of choice in various biomedical applications, including for urinary tract implants [17,18,19]. Parylene coatings offer several advantages in this context. First, its biocompatibility and biostability are essential for urological implants to minimize the risk of adverse reactions or complications [20,21]. Furthermore, parylene coatings exhibit good barrier properties, even at high-temperature exposure, protecting the implant material and preventing potential issues such as the corrosion and degradation of medical devices placed inside the human body [22]. Parylene demonstrates a high resistance to a wide range of chemicals and body fluids, including urine, and shows a low coefficient of friction, that reduces frictional forces and potentially minimizes irritation, discomfort, or damage to the surrounding tissues [23]. Silicon oxide layers have gained well-known recognition for their effective barrier capabilities [24,25,26,27]. However, the attributes of silicon oxide-based compounds extend beyond barrier functions, positioning them as promising coating materials for optical purposes. Silicon oxide-based compounds exhibit remarkable characteristics, such as hardness, transparency, and hydrophilicity [28,29,30,31]. Additionally, significant interest has been directed towards the surface functionalization of silicon oxides. Researchers have studied enhancing surface properties through the deposition of robust overlayers, particularly in the field of optoelectronic device miniaturization [32,33].

The aim of this study is to investigate the effectiveness of two parylene grade coatings, parylene C and VT4, as well as a silicon-based finish coating, ceramic-like SiO_x_, in preventing encrustation for a urinary bladder pressure MEMS sensor system. An in vitro encrustation system was developed and employed to simulate the natural urine flow in the body and quantify the percentage of mass encrustation. Additionally, Fourier transform infrared spectrometry (FTIR) was utilized to analyze the formation of urinary stones. Surface chemical properties, such as the contact angle and surface charge, were characterized to correlate and discuss the results.

## 2. Materials and Methods

### 2.1. Thin Film Deposition

#### 2.1.1. Parylene Deposition

Parylene, commonly referred to as poly-para-xylylene and its derivates, films were deposited using the Gorham process through low-pressure chemical vapor deposition (LPCVD) at room temperature [34]. Parylene films were performed in a CX-30 PC hydride machine, able to deposit inorganic layers by plasma-enhanced chemical vapor deposited (PECVD), provided by Coat-X SA, La Chaux-de-Fonds, Switzerland. To enhance the adhesion of the parylene film to the substrates, an adhesion promoter, methacryloxypropyl trimethoxysilane (Silane A174), was evaporated in the chamber for 3 min prior to the parylene deposition. In order to maintain a constant pressure of 80 µbar in the deposition chamber, a precise control of the sublimation temperature ranging from 80 to 150 °C was employed. The pyrolysis temperature was set at 700 °C to effectively cleave the dimers into monomers. Afterwards, the monomers condensed and polymerized inside the chamber, leading to the creation of a polymeric film at normal room temperature. 

Two different parylene grades, namely parylene C and VT4, were deposited and evaluated. Parylene-based encapsulation solution is considered a very promising alternative to protect biomedical devices. Parylene C, or poly(chloro-*p*-xylylene), is the most commonly used variant of parylene. It demonstrates very low permeability to gases and moisture [22,35]. In addition, parylene C offers improved chemical resistance compared to the basic parylene grade (parylene N), making it suitable for applications that involve exposure to harsh environments. Parylene VT4, poly(tetrafluoro-*p*-xylylene), also categorized as fluorinated parylene or parylene F, incorporates fluorine atoms within its aromatic sites. Fluorinated parylene grades are known to exhibit superior thermal stability and possess a low dielectric constant [22,36]. 

#### 2.1.2. SiO_x_ Finish Coatings

Silicon oxide (SiO_x_) layers were obtained by dissociating a precursor called hexamethyldisiloxane (HMDSO) precursor and oxygen (O_2_) gas molecules using a capacitively-coupled high-frequency plasma at 13.56 MHz. By adjusting the ratio of O_2_ to HMDSO, the wettability properties of the coating layer could be modified. The O_2_/HMDSO ratio allowed for the transformation of the layer’s composition from purely inorganic to a more polymer-like layer due to the incorporation of organic groups from the HMDSO precursor [37]. In this study, a ratio of 10:1 was used for the SiO_x_ coating, similar to the layer developed by Hogg et al. [38].

### 2.2. In Vitro Encrustation Test

Chemically defined synthetic urine was prepared according to Griffith et al. [39]. Solutions A and B, described in Table 1, were prepared with distinct amounts of different salt components (Carl Roth GmbH, Karlsruhe, Germany). Solution A was supplemented with urease (Merck KGaA, Darmstadt, Germany) to induce the crystallization process. The following chemical equations explain the development of struvite (so-called infection stones). Urease triggers the hydrolysis of the urea, producing ammonia. The increase in urinary pH leads to increased precipitation of calcium and magnesium phosphate.
NH_2_-CO-NH_2_ + H_2_O + Urease → CO_2_ + 2NH_3_(1)

Urea is hydrolyzed in the presence of urease.
2NH_3_ + 2H_2_O → 2NH_4_^+^ + 2OH^−^
(2)

Ammonia is hydrolyzed to ammonium ions. Binding with available cations produces magnesium ammonium phosphate (struvite), (NH_4_)Mg[PO_4_]·6H_2_O (pH > 7.5).
2NH_3_ + CO_2_ → CO_2_ + H_2_O → H_2_CO_3_ → 2H^+^ + CO_3_^2−^
(3)

Carbon dioxide is oxidized to bicarbonate. Binding with available cations produces carbonate apatite Ca_10_(PO_4_)6CO_3_(OH)_2_ (pH > 6.8).

The temperature in the incubation system was set to 37 °C. The incubation system simulates the physiological conditions of a urinary bladder. A 500 mL bioreactor is used to simulate the physiological conditions in the urinary bladder. The typical filling volume of a human urinary bladder is estimated at 300 to 500 mL. The urine is well mixed by natural human movements and detrusor contractions. Natural micturition occurs in response to the filling pressure in the urinary bladder and by neuronal reflexes [40,41]. For our investigations, mixing was performed with the help of a magnetic stirrer with 80 revolutions per minute. Every 4 h 250 mL of synthetic urine was exchanged and every 24 h the medium was completely replaced. An experimental period of 7 days was allowed for each study. Catheterization every 4 h is recommended in the urological practice for patients with neurogenic bladder emptying disorders and residual urine in the urinary bladder should be expected. In preparation for a clinical study to analyze encrustations and biofilm on urinary bladder implants, in vitro studies were performed in the research department of Urology, Rostock University Medical Center and the parameters used in this study were determined by Buchholz et al. [42]. The encrustations were quantitatively analyzed by measuring the weight of the silicon wafer samples before and after the experiment. Samples were dried at 37 °C for 3 days to ensure that all residual liquid had evaporated.

#### Fourier Transform Infrared Spectroscopy

Encrustations deposited on silicon wafer samples were dried before the analysis and were ground in an agate mortar. The analysis of the implants was done by polarized light microscopy (BX43 Olympus, Hamburg, Germany) and the Urinary Stone Analyzer Fourier transform infrared spectrometry system FTIR ALPHA (Bruker, Billerica, MA, USA). Spectra were recorded between 2000 and 400 cm^−1^ (resolution 2 cm^−1^) for 120 scans and compared to the OPUS™ reference library, Version 7.5 (Bruker, Billerica, MA, USA). Semi-quantitative results were recorded in percentages. The qualitative analysis was based on the comparison of the spectrum of a sample with the spectrum of the reference material.

### 2.3. Contact Angle

Surface wettability was determined by the sessile drop method using the Drop Shape Analyzer—DSA25 (Krüss, Hamburg, Germany) [43]. One μL drop of distilled water and synthetic urine (pH 6.4; for formulation see Section 2.2) were deposited on the silicon wafer sample surface (*n* = 2 at 5 drops each). Distilled water and synthetic urine contact angle values were calculated with the supplied software (ADVANCE, V.1.7.2.1, Krüss, Hamburg, Germany) via Young’s equation and the surface free energy according to Owens, Wendt, Rabel, and Kaelble (OWRK).

### 2.4. Surface Charge

Zeta potential measurements were conducted on wafer samples (2 × 1 cm) using the SurPASS system (Anton Paar, Ostfildern, Germany) to determine the surface charge. Silicon wafer samples were mounted with a gap height of 100 µm between the two samples. The measurements were performed in 0.001 mol/L KCl solution ranging from pH 5.0 to 8.0. The streaming current was determined depending on the pressure (max. 400 mbar). Finally, the zeta potential was calculated according to the method of Helmholtz-Smoluchowski. Measurements on each surface were performed in quadruplicate on three independent sample pairs.

## 3. Results

### 3.1. In Vitro Encrustation Test

The formation of urinary stones in vivo depends on a variety of factors, such as pH, protein excretion, the presence of foreign material, bacterial infections, or dietary habits. Therefore, in vitro encrustation models have been developed to enable systematic studies that could contribute to a deeper understanding of the problem. We built our in vitro encrustation system based on this knowledge and used synthetic urine introduced by Griffith [39]. In this approach, two different unsaturated solutions were pumped into a bioreactor in which the polymer samples were placed. The initial pH of the synthetic urine was 5.5 and over time, the urease decomposed the urea into carbon dioxide and ammonia. This caused the pH to shift to the alkaline range, resulting in the precipitation of salt crystals. The rate of pH increase was caused by the urease reaction. The pH value reached the equilibrium state at a pH of 8.9. The encrustation mass was normalized, and each group comprised 4 samples. The mass of the samples with identical surface areas was measured before and after synthetic urine immersion. The normal distribution within each group was assessed using the Shapiro-Wilk test [44]. To compare the data groups, a two-tailed, unpaired Welch’s *t*-test was performed, revealing statistically significant differences (*p* < 0.05). The data analysis was conducted using R software version 4.2.2 [45,46].

Figure 1 illustrates the results of the encrustation test, indicating the encrustation mass (mg) measured. The data is categorized based on different parylene grades and further classified according to the type of finish coating applied. For parylene C, the mean value of the encrustation mass is measured at 9.55 mg. In the case of parylene VT4, the encrustation decreases to 7.5 mg when the parylene comes into direct contact with synthetic urine (“No Finish” group). When parylene is covered with a SiO_x_ finish coating, the encrustation levels decrease further to 5.23 mg and 4.15 mg for parylene C and VT4, respectively.

#### Fourier Transform Infrared Spectroscopy

Encrustations are mostly mixtures of substances. Depending on the excited vibrational and rotational states of the molecules, characteristic absorptions were measured using Fourier transform infrared spectrometry. For an accurate qualitative and quantitative analysis, suitable reference substances and mixtures must be available (OPUSTM library, Bruker Optik GmbH, Ettlingen, Germany). The qualitative analysis was based on the comparison of the spectrum of a sample with the spectrum of the reference material. The measured absorption spectra of the mineral deposits are illustrated by the blue curve, while the absorption spectra of the OPUS reference library are shown in red.

FTIR analysis of the encrustations confirms that calcium oxalate (whewellite and weddellite), phosphate crystals (struvite, brushite, and apatite), and protein are major constituents and correlate well with the studies on human implants [47,48,49]. The FTIR analysis of urine encrustation depicted by the blue curve in Figure 2 consists of ammonium magnesium phosphate hexahydrate (struvite), carbonate apatite (dahlite), and calcium oxalate dihydrate (weddellite), with a compound ratio of 70, 20, and 10%, respectively. The hit quality with respect to the reference curve, red curve, is 968.

### 3.2. Contact Angle

The contact angle serves as a measure to characterize the hydrophilicity or hydrophobicity of a material. Typically, materials with a contact angle of <90° are labeled as hydrophilic, while those with a contact angle >90° are considered hydrophobic [50]. To closely emulate the in vivo conditions and to consider the potential impact of the solution on surface wettability, contact angle measurements using synthetic urine (hatched columns), as described earlier, and distilled water for reference measurements were conducted. Illustrated in Figure 3, the findings indicate that parylene C exhibits a lower water contact angle compared to parylene VT4, 91.9° (±2.1) and 94.7° (±1.6), respectively. This distinction is also observed when synthetic urine is used as the test liquid, 86.7° (±1.8) for parylene C and 95.1° (±2.7) for parylene VT4. Although the values obtained with synthetic urine differ slightly from those obtained with distilled water, both solutions confirm that parylene C possesses a more hydrophilic surface compared to parylene VT4. Furthermore, the application of a finish SiO_x_ coating leads to significantly lower contact angle values of 56.7° (±3.6) and 54.1° (±2.7) for distilled water and synthetic urine, respectively. It has to be noted that the SiO_x_ layer contact angle changed after several days of water immersion stabilizing at 71.5° (±2.2).

### 3.3. Surface Charge

The Zeta potential of the coated silicon wafer sample surfaces was measured in a pH range from 5.5 to 7.5 (see Appendix A, Figure A1). Specifically, the zeta potential value at pH 6.4, representing the average urine pH of healthy humans, was then extracted from the measurements. These results are presented and illustrated in Figure 4 for parylene C, VT4 films, and SiO_x_ coating.

Zeta potential measurements detected negative charges for all examined materials. As depicted in Figure 4, within the characteristic pH value of 6.4, the parylene C coating demonstrates a more negative value of −65.0 mV (±3.3) when compared to parylene VT4, which displays −55.7 mV (±3.6). The SiO_x_ coating exhibits a charge value of −51.9 mV (±3), making it the least negatively charged sample among all tested coatings. The complete data set of zeta potential measurements across the pH range of 5.5–7.5 are presented in Appendix A.

## 4. Discussion

As demonstrated in former studies, parylene-based encapsulation, and in particular parylene C grade, is considered a well-adapted solution for biomedical applications [20,21,23,51,52]. Implantation of biomaterials into the urinary tract is disadvantaged by stone formation, bacterial adherence and, ultimately, encrustation through biofilm formation resulting from a multifactorial disturbance of the delicate balance between numerous physico-chemical and biochemical processes [53,54]. Non-infectious stone formation and encrustation usually result from metabolic imbalances, often on the tubular level. In contrast, infectious stone formation and biofilm-induced encrustation are linked to the enzymatic activity of bacteria. The best known are urease-producing species, such as proteus mirabilis, which increase the pH of the urine. This alkalization, in turn, decreases the solubility of urinary calcium and magnesium salts and thus facilitates the encrustation [1,55]. From the implant functionality perspective, encrustation can lead to reduced sensitivity, measurement drift, implant dysfunction, or complete implant failure [56]. As the encrustation builds up, it can create a barrier between the implant and its surrounding environment, introducing various complications. In our case, the sensitivity of the MEMS sensors used for monitoring pressure can be impacted negatively due to the encrustation. When encrusted, the implant’s sensing capabilities may be altered or distorted. This reduction in sensitivity can cause inaccurate readings or measurements. Encrustation can also lead to measurement drifts or fluctuations caused by the alteration of the encrusted implant surface. Consequently, the reliability and accuracy of the measurements may deteriorate, making it challenging to obtain consistent data for long-term analysis or monitoring purposes [14].

The first phase of our investigations involved conducting urinary encrustation tests to evaluate the efficacy of different parylene films. Our aim was to use a reproducible in vitro system that simulates the in vivo infection situation realistically. To ensure this, we used defined amounts of urease instead of adding urease-producing bacteria. Furthermore, the quality and quantity of encrustations depend on the protein amount, which is why we decided to optimize our system and use albumin in a concentration of 333 mg/L, resembling a macroalbuminuria in our in vitro system [57]. Urinary encrustations consist of a mixture of crystals. In clinical practice, according to the guidelines for diagnosis, therapy, and metaphylaxis of urolithiasis, FTIR is recommended as a method for the urinary analysis of encrustations. The comparison of the analyzed crystal compounds with real combinations of urinary stones ensured the presence of urinary encrustation on tested samples. Among the parylene tested, grade C exhibited slightly less efficiency in preventing encrustation compared to grade VT4. However, when compared to other polymeric materials, parylene films present excellent results. Rebl et al. [58] investigated different polymers in a systematic screening approach to determine which surface parameter properties indicate low encrustation properties. The study showed that the most promising tested candidates for preventing encrustation on ureteral stents were a thermoplastic urethane, Elastollan 1185A 10 FC and a styrene-butadiene copolymer, Styroflex 2G 66, with 2.4% and 2.1% of mass encrustation, respectively. In comparison, parylene C and VT4 only exhibited an encrustation mass of 9.55 mg and 7.5 mg, which corresponds to a percentage of 2.2% and 1.7%, respectively, making them attractive options for encrustation prevention. Particularly, parylene-based encapsulation solutions prove to be an outstanding choice due to their properties, such as excellent barrier performance, thermal stability, pinhole-free coating, and biocompatibility. Furthermore, we observed that the addition of a SiO_x_ layer to the coatings significantly improved their efficiency in preventing encrustation. This finding suggests that surface coatings might hold the key to enhancing encrustation prevention even further, reaching 5.23 mg and 4.15 mg of encrustation mass when parylene C and VT4, respectively, are covered with a finish coating of SiO_x_. Considering that the hydrophobicity of a surface can significantly influence interactions with surrounding fluids and substances, we next explored the contact angles of the coatings. Both parylene C and VT4 displayed comparable contact angles, indicating a slight hydrophobic nature. On the other hand, SiO_x_ showed a notably strong hydrophilic character. Lastly, we analyzed the zeta potential of the coatings, which is a crucial factor affecting surface charge and interactions with surrounding substances. Parylene C exhibited the most negatively charged surface, possessing a zeta potential of −65 mV. In contrast, both parylene VT4 film and SiO_x_ coating displayed similar zeta potential values, ranging from 55 to 52 mV. Interestingly, we discovered that the zeta potential alone does not entirely explain the encrustation results. While parylene C demonstrated a strong negative zeta potential, indicating heightened repulsion forces, it did not directly correlate with more effective prevention of encrustation. This finding partially aligns with the assumption discussed by Rebl et al. [58], suggesting that a negative surface charge of about −60 mV was the most suitable for use as stent materials, as the deposition of crystalline biofilms was minimized. It could be hypothesized that an excessively negative surface (inferior to −60 mV) may not be advantageous in preventing encrustation. Furthermore, our investigations into zeta potential and contact angle indicate that hydrophobicity might play a more essential role than zeta potential in explaining the differences in encrustation results among the coatings. This leads us to consider other chemical and physical surface properties that are potential contributing factors in the encrustation prevention mechanism. Additionally, the results revealed that the interaction of various surface properties, rather than individual factors, may hold the key to understanding and optimizing encrustation prevention. Further developments and characterizations have to be conducted to explore the exact mechanisms governing encrustation. 

When considering encapsulation materials for in vivo implants or sensors, it was found that biocompatible parylene films alone demonstrate interesting barrier properties, even when in direct contact with fluids [59]. However, the barrier performance can be improved by several orders of magnitude once parylene is combined with inorganic thin films to build up multilayer structures [25,38,60,61]. A multilayer parylene-based barrier can prevent the penetration of moisture, gases, chemicals, and other contaminants that can degrade sensor performance for long-term implantation. Moreover, parylene-based encapsulation, deposited through a vapor-phase process, results in a uniform, conformal and pinhole-free coating. It can cover high-aspect ratio geometries, ensuring uniform encapsulation of 3D devices. This capability is particularly beneficial for MEMS sensors, which often have complex microstructures that need protection. Furthermore, the parylene-based solution can as well be considered an optimal solution for electronic sensors that may be exposed to a wide range of operating temperatures, due to its excellent thermal stability [22,62], combined with effective electrical insulator properties [36,63].

## 5. Conclusions

In conclusion, our investigations in encrustation prevention and the properties of parylene thin films highlighted the suitability of parylene as an encapsulation solution for biomedical and, more precisely, for urological applications. Parylene-based coatings, with their barrier properties, conformal deposition capability, thermal stability, biocompatibility, and electrical insulation, prove to be very promising candidates for sensor protection. The study demonstrated that parylene films can effectively protect urological MEMS sensors from adverse effects, and the SiO_x_ finish coating additionally enhances the efficiency of the parylene-based encapsulation in preventing encrustation, suggesting the potential of composite coatings to optimize sensor protection even further. Understanding to a certain level the interaction of surface properties, such as surface charge and wettability, contributes to improving the development of surface treatments or finish coatings for encrustation prevention. Finally, a parylene-based solution stands out for its exceptional intrinsic properties, making it an outstanding choice for long-term urological implant encapsulation and effectively preventing encrustation.

## Figures and Tables

**Figure 1 polymers-15-03559-f001:**
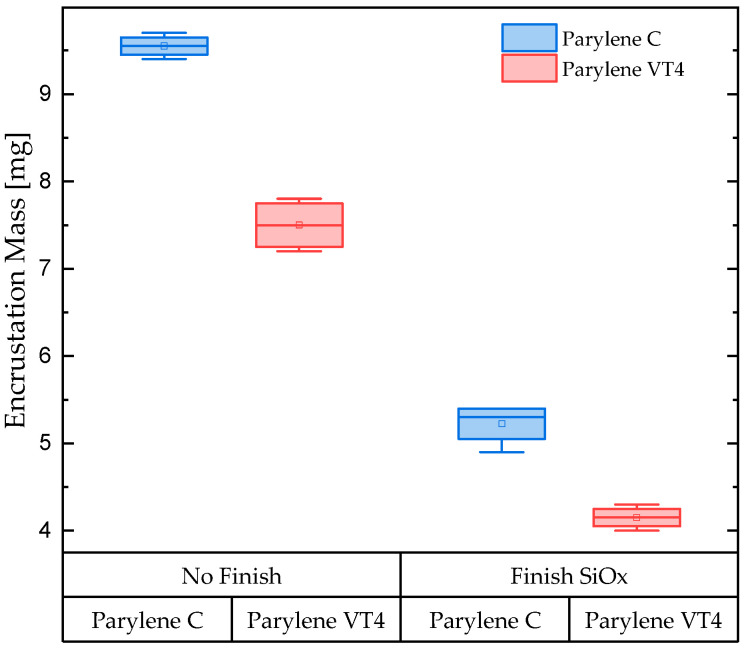
Urine encrustation mass (mg) for the coating groups parylene C and parylene VT4 without any finish coating (No Finish) and with a finish ceramic-like SiO_x_ coating (Finish SiO_x_).

**Figure 2 polymers-15-03559-f002:**
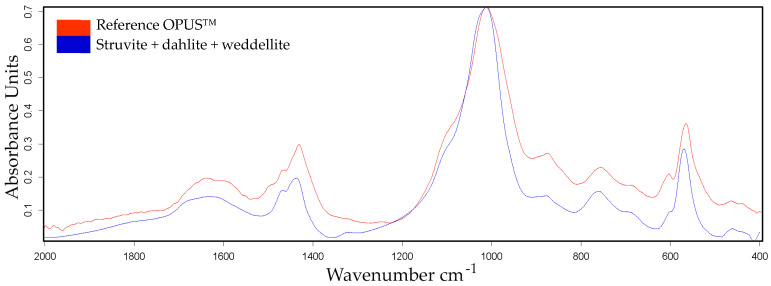
FTIR analysis of urine encrustations on silicon wafer samples. The measured absorption spectra and the absorption spectra of the OPUS™ reference library are illustrated by the blue and red curves, respectively. The encrustation consists of ammonium magnesium phosphate hexahydrate (struvite), carbonate apatite (dahlite), and calcium oxalate dihydrate (weddellite) with a compound ratio of 70, 20, and 10%, respectively.

**Figure 3 polymers-15-03559-f003:**
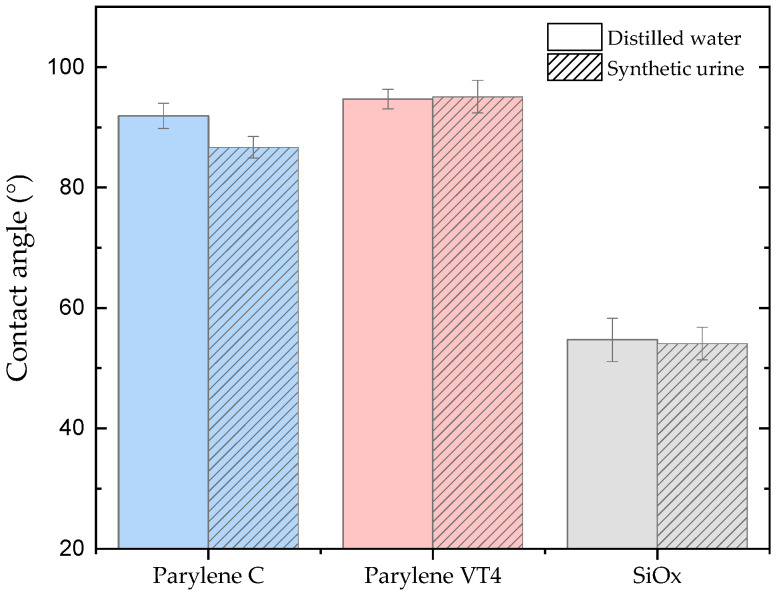
Contact angle measurements of parylene C, VT4, and SiO_x_ coating for distilled water and synthetic urine (hatched columns) on silicon wafer samples. Finish SiO_x_ coating demonstrates strong hydrophilic properties compared to both parylene-grade films.

**Figure 4 polymers-15-03559-f004:**
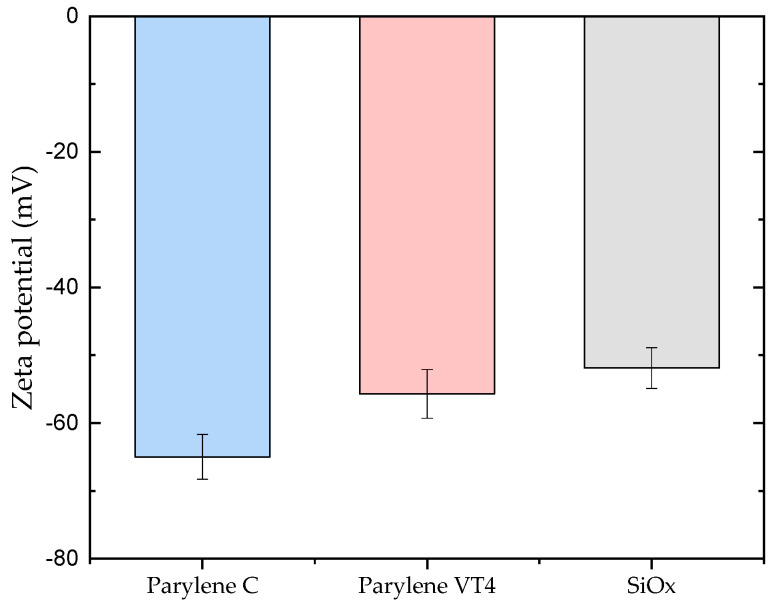
Zeta potential measurements of parylene C, VT4, and SiO_x_ coating at a pH value of 6.4, representing the average urine pH of healthy humans.

**Table 1 polymers-15-03559-t001:** Solution A and B, in vitro encrustation system.

Solution A	Solution B
calcium chloride-dihydrate	urea
magnesium chloride-hexahydrate	potassium-dihydrogenphosphate
sodium citrate	creatinine
sodium chloride	ammonium chloride
sodium sulfate	albumin
potassium chloride	sodium oxalate
urease, type IX	

## Data Availability

The data presented in this study are available on request from the corresponding author.

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
