# Peer review of "An Evaluation of Parylene Thin Films to Prevent Encrustation for a Urinary Bladder Pressure MEMS Sensor System"

_polymers, 2023, doi:10.3390/polym15173559_

Round 1

Reviewer 1 Report

Authors have studied the relative performance of parylene  C and parylene VT4 grade coating in preventing the encrustation in urethral implants e.g, silicon based MEMS sensor. Both the material demonstrate superior results in prevention of encrustation. Adding SiOx coating enhances the performance even further by reducing the contact angle. Various methods are used for chemical and physical analysis of the urinary stones.

A well written article with clearly described method. Parylene C being a preferred material for coating MEMS implants is nothing new. SiOOx coating seems to enhance the performance. But the relative superior performance of parylene VT4 might just because of the way results are analyzed (see comment 2). 

Following are the comments/suggestion to make the article better,

Major comments:

1. In section 2.2 authors have mentioned the different parameters used for the incubation system e.g, size of the bio reactor is 500ml, mixing done with 800 rev per minute, 250ml of synthetic urine exchanged every 4 hours and so on. It will be good if author can add some justification behind the numerical value of these parameters.

2. In section 3.1 and figure-1 the results are presented in 'mass percentage (%) for the coating groups '. There needs to have some justification on using this parameter since the mass of the repeating group in Parylene C and parylene VT4 are not same. Parylene VT-4 might have undue advantage for having a heavier repeating unit in comparison to parylene C.

3. Authors have used FTIR for chemical analysis of the stone that formed and the chemical signature resembles with the reference stone found in human implants. What does that signifies?

4. Adding few sentences on why parylene VT4 is used to compare with parylene C. There are many other parylene compound generated by substituting the -Cl group of parylene C by e.g, -OH, -CHO, -COOH.

Minor comments:

5. Some references are missing. E.g, in section 3.1, Shapiro-Wilk test, R Software and few others.

6. "Parylene", sometimes written with capital 'P' and sometimes small. Try to be consistent.

Author Response

Dear Reviewer, 

Please find attached the report with the answers of the authors. 

Best regards, 

Sébastien Buchwalder

Reviewer 2 Report

This manuscript report pressure sensor based on materials. The mansucript can be accpeted after the following issues are solved.

1. What is the main question addressed by the research? Please highlight the noverlty to discribe.

2. Why ureteral pressure sensor new? Do you consider the topic original or relevant in the field? 

3. What does it to the subject area compared with other published

material?

4. What specific improvements should the authors consider regarding the

methodology? 

5. Are the conclusions consistent with the evidence and arguments presented and do they address the main question posed?

6. The authors can cite more functional materials or coated materials papers to support the background.

7. What is the purpose of Figure 2?

8, Any news for parylene C, VT4 and SiOx?Why choose these three to compare

The language is fine for readers

Author Response

(The authors gave the same response as above.)

Round 2

Reviewer 2 Report

The authors did answer all my issues. I have no more comments

It is fine for me